**Comment**

# Psychology should move from selective allyship to empowered actions to tackle global crises

Maja Kutlaca, Helena R. M. Radke & Özden Melis Uluğ

Psychology is committed to the principle of nonmaleficence (i.e., do no harm). Yet the discipline's past failures and its current selective allyship with only some crises paint a problematic picture. The authors argue that psychology as a discipline and psychological associations as its representatives should better uphold their ethical responsibility.

The world faces a series of overlapping global crises including ongoing wars and genocides, widespread disasters due to climate change, as well as political instability and the systemic targeting of marginalized and oppressed groups. Psychology as a discipline is well-placed to respond to crises and suffering. For example, psychological science provides us with the tools required to understand and mitigate conflict and build peace, support individuals and groups who are affected by such crises, foster solidarity between groups, and promote pro-environmental behaviors.

Central to psychology's ethical framework is the principle of nonmaleficence or the duty to do no harm, which, like medicine, obliges psychologists to not cause harm and actively prevent harm where possible[1]. This principle traces its roots to the Nuremberg Code, which outlined a set of ethical standards in response to the atrocities committed during World War II. As a result, the American Psychological Association (APA) established the first professional code of ethics, a model which has been adopted by other psychological associations around the world (e.g., national psychological associations in Canada, Norway, and France)[1].

Despite these commendable contributions, it is important to acknowledge that psychology has also been complicit in many harmful practices. Historically, this includes ethically questionable experiments on children (e.g., Little Albert Experiment), adults (e.g., Stanford prison experiment, Milgram studies), and animals (e.g., Harlow's experiments on monkeys), as well as reinforcing discrimination based on race, ethnicity, disability, sex, gender, and sexual orientation through research, teaching, and practice[2].

## Psychology's ethical responsibilities regarding social issues

The failure to uphold the principle of nonmaleficence extends beyond research practices. Even though psychology as a discipline is aligned with the Helsinki accords and committed to preventing discrimination and ensuring human dignity, historical records indicate that the discipline has often failed to uphold and sometimes even violated this principle when addressing broader societal issues. For example, in psychiatry, the American Psychiatric

Association banned its members from taking part in torture and coercive interrogations carried out by the U.S. government in the aftermath of the September 11 attacks (https://www.psychiatry.org/getattachment/015a4fab-0de9-4c57-96b3-126c38e40e48/Position-Psychiatric-Participation-in-Interrogation-of-Detainees.pdf). In contrast, the American Psychological Association (APA) failed to prohibit members to desist from participating in these practices according to the Hoffman Report (https://www.apa.org/independent-review/revised-report.pdf). Worryingly, members' concerns raised over these problematic practices were ignored by the APA leadership at the time[3]. It was only after the release of the Hoffman Report in 2015, which brought to light the APA's involvement, that the APA formally apologized and prohibited its members from being involved in these practices (https://www.apa.org/news/press/releases/2015/07/independent-review-release).

Fortunately, in the past decade, the discipline has taken important steps to uphold the principle of nonmaleficence and to amend its previous ethical shortcomings. Some psychological associations have acknowledged the discipline's problematic involvement in human rights violations and issued formal apologies for the discipline's role in perpetuating racism (APA's Apology to People of Color: https://www.apa.org/about/policy/racism-apology), erasing Indigenous rights (Australian Psychological Association Apology to Aboriginal and Torres Strait Islander People: https://psychology.org.au/community/reconciliation-and-the-aps/aps-apology), and discriminating against LGBTQIA+ communities (Society for Psychoanalysis and Psychoanalytic Psychology, Division 39 APA: https://div39members.wildapricot.org/LGBTQ-Apology), and have explicitly committed to combating prejudice and discrimination. Other national and international psychological associations have made statements addressing important societal issues (British Psychological Society Statement on Racial Injustice: https://www.bps.org.uk/news/bps-statement-racial-injustice) and have institutionalized anti-discrimination practices within their codes of ethics[4–6]. These developments, particularly those that move beyond symbolic acknowledgments of past harms toward the advancement of human rights, suggest that the discipline is beginning to critically engage with and redress its historical shortcomings.

## Doing harm through selective allyship

We argue that the violation of the principle of nonmaleficence continues today in the form of selective allyship. We characterize selective allyship as the inconsistent application of the principle of nonmaleficence, which results in acknowledging and preventing the suffering of some social groups while remaining silent, denying, or actively justifying the suffering of other social groups. Following the full-scale Russian invasion of Ukraine in February 2022, a number of psychological associations rightly demonstrated their allyship with Ukrainian people almost immediately after the Russian invasion by issuing public statements, sharing

resources, organizing training workshops and webinars, and hosting displaced academics (https://humanrightspsychology.org/content-areas/war-and-human-rights/ukraine/). These organizations include, but are not limited to, the APA, British Psychological Society, Canadian Psychological Association, European Federation of Psychologists' Associations, and German Psychological Society.

However, there are conflicts, wars, and suffering all around the world, including the civil war in Myanmar (https://www.theguardian.com/global-development/2025/jan/31/why-is-myanmar-embroiled-in-conflict), the ongoing conflict in Sudan (https://www.bbc.com/news/world-africa-59035053), and the Taliban takeover and ensuing crackdown on human (and especially women's) rights in Afghanistan (https://news.un.org/en/story/2025/05/1162826). We maintain that these conflicts have received little or no attention from psychological associations. More prominently discussed, in response to the Israeli occupation of Palestinian territories and the ongoing displacement, destruction, and harm perpetrated against the Palestinian people in Gaza (which the International Court of Justice stated may amount to a plausible genocide: https://www.icj-cij.org/node/203447), and in the West Bank, psychological associations either responded with significant delay or have yet to position themselves as universal allies[7]. Similar conclusions have been drawn by health professionals who highlighted the unwillingness to speak out about the situation in Gaza and the selective silence of health and social science associations, calling on these organizations to officially recognize the genocide in Gaza[8].

As researchers who study allyship and lead an international network of scholars working on collective action and solidarity, we contend that people are more likely to support others whom they perceive as similar to them or with whom they share economic and political interests. In contrast, they are more likely to avoid those who may question their privileges or remind them of past and present injustices[9]. This can have far-reaching implications for society at large, because support from powerful scientific and professional organizations can shape who in society is seen as worthy of support and who is not. In doing so, selective allyship by professional organizations plays an active role in allowing harm that could otherwise be prevented.

### Looking forward: what can associations and psychologists do to address global crises?

We therefore call on our associations to return to the core principle of doing no harm, which underpins the discipline's code of ethics around the world in a way that promotes proactive, engaged, consistent, long-term, and concrete support for those affected by global crises. This is possible with many promising examples of advocacy for social justice and change both within (e.g., APA's Division 48 Society for the Study of Peace, Conflict and Violence; Australian Psychological Association's Psychologists for Peace Interest Group) and outside of larger psychology associations (e.g., Association of Black Psychologists in the United States: https://abpsi.org/; Association of Psychologists for Social Solidarity in Turkey: https://todap.org/eng/who_is_todap.aspx?link=2). For example, APA's Division 48 called for justice and accountability to confront the atrocities against Palestinian lives and grounded their call in a long-term commitment to justice, psychological healing, and human dignity by imploring psychologists to move beyond silence or neutrality in the face of this atrocity (https://peacepsychology.org/psychologists-call-for-justice-and-accountability). By building on these promising examples and research on allyship, we outline some recommendations for how associations can better address global crises:

1. *Protect All Life and Do No Harm*. Many associations already have ethics committees or at least representatives who should ensure that there is a consistent and unbiased stance taken against all perpetrators, whilst affirming the equal worth of all life. These ethics committees should ensure that all members adhere to the principle that 'no life is more valuable than another.' We encourage our associations to reaffirm their commitment to nonmaleficence and use this principle as a lens through which they undertake all activities on behalf of the discipline.

2. *Mobilize Resources and Show Genuine Allyship*. Associations should offer meaningful support to all affected colleagues and communities and extend this support beyond academia, in solidarity with everyone facing crisis or injustice. Support may come in various forms including but not limited to publishing statements, designing educational programs, providing mental health initiatives, and providing resources.

3. *Engage in Public Education and Advocacy*. Associations need to actively engage in public education and political advocacy. Scientific associations serve as important societal role models and can use their platforms to inform the public and influence policymakers. Given that associations set scientific and professional standards for discipline, they are best placed to engage in advocacy rather than individual colleagues who may face greater social and political risks and are more likely to be silenced.

4. *Commit to Structural Change and Engage in a Democratic Dialogue*. Professional associations must practice humility by acknowledging past and present failures and commit to structural changes to battle all forms of harm. In some cases, acknowledging failure and injustice may lead to conflicts within associations. When those occur, associations need to create spaces for open dialogue with members, including those who are critical of their stance or (in)actions, and must avoid punishing, silencing or ignoring dissenting opinions.

Many of the positive examples of upholding the principle of nonmaleficence have been achieved through the courageous work of our colleagues who have voiced their concerns and put pressure on the associations to change, despite large costs for their careers, reputations, and psychological well-being (e.g., Hoffman report highlighted APA's leadership poor treatment of members who voiced concerns over involvement in torture)[5]. Drawing on research into prejudice confrontation and collective action, we know that exposing injustice is costly for individuals, but the power to challenge existing structures lies in collective numbers. For example, the criticism from over 3000 health professionals led major public health associations to issue a statement acknowledging the genocide in Gaza[8]. Thus, we encourage all psychology researchers and practitioners to mobilize and challenge associations to uphold their code of ethics.

We therefore find ourselves at a crossroads. We call on our discipline to embrace true democratic and inclusive engagement that requires transparency, accountability, and the courage to listen and act, even when it is uncomfortable. Ensuring that we uphold the principle of nonmaleficence, actively prevent harm where possible, and avoid selective allyship is both necessary and a key contribution our discipline can make in these unprecedented times.

**Maja Kutlaca** [1] ✉, **Helena R. M. Radke** [2] **& Özden Melis Uluğ** [3]
¹Psychology Department, Durham University, Durham, UK. ²Psychology Department, James Cook University, Townsville, QLD, Australia. ³School of Psychology, University of Sussex, Brighton, UK.
✉e-mail: maja.kutlaca@durham.ac.uk

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

## Author contributions

M.K.: conceptualisation, writing – original draft, writing – review and editing. H.R.: conceptualisation, writing – original draft, writing – review and editing. O.M.U.: conceptualisation, writing – original draft, writing – review and editing. The authors contributed equally to the work and are listed in alphabetical order.

## Competing interests

Özden Melis Uluğ is a member of APA Division 48. Maja Kutlaca and Helena R. M. Radke declared no competing interests.

## Additional information

**Peer review information** Primary handling editor: Troby Ka-Yan Lui. A peer review file is available.

