## [Transparent Peer Review file · Communications Psychology]

Psychology should move from selective allyship to empowered actions to tackle global crises

Corresponding Author: Dr Maja Kutlaca

Version 0:

Decision Letter:

Dear Dr Kutlaca,

Thank you for your patience during the editorial evaluation. We find your Comment "Professional Ethics on Trial: Psychology's Slow, Reactive, and Selective Allyship Toward Global Crises" of interest but have a number of editorial concerns about the presentation and persuasiveness of the piece. We are interested in the possibility of publishing the Comment in *Communications Psychology*. Still, we would like to consider your responses to these concerns and assess a revised manuscript before we make a final decision on publication.

To aid you with the revisions, I have included a marked-up version of your manuscript, which lays out high-level structural issues with the presentation. To consider a revised version, we require these to be addressed.

EDITORIAL POLICIES AND FORMATTING

* **TRANSPARENT PEER REVIEW:** *Communications Psychology* uses a transparent peer review system. This means that we publish the editorial decision letters including Reviewers' comments to the authors and the author rebuttal letters online as a supplementary peer review file. We publish these records for all accepted manuscripts. However, on author request, confidential information and data can be removed from the published reviewer reports and rebuttal letters prior to publication. If your manuscript has been previously reviewed at another journal, those Reviewers' comments would not form part of the published peer review file.

If you have any questions about any of our policies or formatting, please don't hesitate to contact me.

Please use the following link to submit your revised manuscript:

Link Redacted

We hope to receive your revised paper within 4 weeks; please let us know if you aren't able to submit it within this time so that we can discuss how best to proceed. If we don't hear from you, and the revision process takes significantly longer, we may close your file.

We understand that due to the current global situation, the time required for revision may be longer than usual. We would appreciate it if you could keep us informed about an estimated timescale for resubmission, to facilitate our planning. Of course, if you are unable to estimate, we are happy to accommodate necessary extensions nevertheless.

Please do not hesitate to contact me if you have any questions or would like to discuss these revisions further. We look forward to seeing the revised manuscript and thank you for the opportunity to review your work.

Best regards,

Troby Lui, PhD
Associate Editor
Communications Psychology

If you experience problems in linking your ORCID, please contact the Platform Support Helpdesk.

Version 1:

Decision Letter:

** Please ensure you delete the link to your author homepage in this e-mail if you wish to forward it to your co-authors **

Dear Dr Kutlaca,

Your Comment titled "From Selective Allyship to Empowered Action: What Psychology as a Discipline Can Do to Tackle Global Crises" has now been assessed editorially. Thank you for your comprehensive changes in response to the initial 'Revise' decision. I am delighted to say that we are happy, in principle, to publish it in Communications Psychology.

If the revised paper is in Communications Psychology format, in an accessible style, and of appropriate length, we shall accept it for publication immediately. I have attached an edited version of your manuscript and ask you to attend to each comment in detail. The edits on the current version provide instructions for the appropriate referencing style for Comments and stylistic guidance about the format, especially regarding brevity and clarity. We also highlight passages that require revisions to ensure that stated facts are described with precision and appropriately supported by references, and that the narration aligns with the nature of a Comment, i.e. an opinion piece.

EDITORIAL REQUESTS:

* Please review the changes in the attached copy of your manuscript, which has been edited for style, and address the comments and queries I have added. If you provide a tracked-changes version in revision, please implement all of our edits and delete all editorial comments, first.

*If you have not done so already, please alert me to any related manuscripts from your group that are under consideration or in press at other journals, or are being written up for submission to other journals (see www.nature.com/authors/editorial_policies/duplicate.html for details).

FORMATTING GUIDELINES:

You will find a complete list of formatting requirements following this link: <https://www.nature.com/documents/commsj-style-formatting-checklist-comment.pdf>

Please use the checklist to prepare your manuscript for final submission. In the following, I also highlight some issues of particular importance.

** Title

Titles should be descriptive of the main message your manuscript conveys and should not exceed 90 characters (including spaces). Please note that punctuation is not allowed, nor are titles of the following format: "title: subtitle". Although the choice of title is largely yours, may I suggest the following: Psychology should move from selective allyship to empowered actions to tackle global crises

** Length

The ideal length for Comment article in Communications Psychology is 1,500 words. We have some flexibility, however, please ensure the revision is not longer than the current version.

* References

There is a strict upper limit of 15 References for a Comment.

References appear as superscript Arabic numerals, in order of mention. The reference list mentions references in the numerical order in which they are mentioned in the main text. If a reference is cited more than once, the same number is used throughout the text and the reference receives a single entry in the reference list.

Only papers that have been published or accepted by a named publication should be in the reference list (preprints and citations of datasets are also permitted). Unpublished/Submitted research should not be included in the reference list; it should only be mentioned briefly and parenthetically in the main text. Note that no major arguments should rely on unpublished research. For records that appeared on websites, but not within academic journals or other forms of publishing, please include a hyperlink to the PDF instead of including them in the reference.

Footnotes are not used.

* Competing interests

Please include a "Competing interests" statement after the References. Note that we ask authors to declare both financial and non-financial competing interests. For more details, see <https://www.nature.com/authors/policies/competing.html>. If you have no financial or non-financial competing interests, please state so: "The authors declare no competing interests."

SUBMISSION INFORMATION:

In order to accept your paper, we require the following:

* A cover letter describing your response to our editorial requests.

* The final version of your text as a Word or TeX/LaTeX file, with any tables prepared using the Table menu in Word or the table environment in TeX/LaTeX and using the 'track changes' feature in Word.

Communications Psychology is a fully open access journal. Articles are made freely accessible on publication. For further information about article processing charges, open access funding, and advice and support from Nature Research, please visit <https://www.nature.com/commpsychol/open-access>

Please note that your paper cannot be sent for typesetting to our production team until we have received this information; **therefore, please ensure that you have this ready when submitting the final version of your manuscript.**

ORCID

Communications Psychology is committed to improving transparency in authorship. As part of our efforts in this direction, we are now requesting that all authors identified as 'corresponding author' create and link their Open Researcher and Contributor Identifier (ORCID) with their account on the Manuscript Tracking System (MTS) prior to acceptance. ORCID helps the scientific community achieve unambiguous attribution of all scholarly contributions. For more information please visit <http://www.springernature.com/orcid>

For all corresponding authors listed on the manuscript, please follow the instructions in the link below to link your ORCID to your account on our MTS before submitting the final version of the manuscript. If you do not yet have an ORCID you will be able to create one in minutes.

IMPORTANT: All authors identified as 'corresponding author' on the manuscript must follow these instructions. Non-corresponding authors do not have to link their ORCIDs but are encouraged to do so. Please note that it will not be possible to add/modify ORCIDs at proof. Thus, if they wish to have their ORCID added to the paper they must also follow the above procedure prior to acceptance.

To support ORCID's aims, we only allow a single ORCID identifier to be attached to one account. If you have any issues attaching an ORCID identifier to your MTS account, please contact the [Platform Support Helpdesk](http://platformsupport.nature.com/).

Link Redacted

We hope to hear from you within two weeks; please let us know if the process may take longer.

Best regards,

Troy Lui

Troy Lui, PhD
Associate Editor
Communications Psychology
